# Characterization and Localization of Sol g 2.1 Protein from *Solenopsis geminata* Fire Ant Venom in the Central Nervous System of Injected Crickets (*Acheta domestica*)

**DOI:** 10.3390/ijms241914814

**Published:** 2023-10-01

**Authors:** Siriporn Nonkhwao, Prapenpuksiri Rungsa, Hathairat Buraphaka, Sompong Klaynongsruang, Jureerut Daduang, Napamanee Kornthong, Sakda Daduang

**Affiliations:** 1Faculty of Pharmaceutical Sciences, Khon Kaen University, Khon Kaen 40002, Thailand; siriphorn_nonkhaow@kkumail.com (S.N.); prapen_rungsa@kkumail.com (P.R.); hathairat_s@kkumail.com (H.B.); 2Protein and Proteomics Research Center for Commercial and Industrial Purposes (ProCCI), Khon Kaen University, Khon Kaen 40002, Thailand; somkly@kku.ac.th; 3Faculty of Associated Medical Sciences, Khon Kaen University, Khon Kaen 40002, Thailand; jurpoo@kku.ac.th; 4Chulabhorn International College of Medicine, Thammasat University, Pathumthani 12120, Thailand; napamaneenatt@gmail.com

**Keywords:** *Solenopsis geminata*, recombinant protein, Sol g 2.1 protein, insect central nervous system, octopamine receptors

## Abstract

*Solenopsis geminata* is recognized for containing the allergenic proteins Sol g 1, 2, 3, and 4 in its venom. Remarkably, Sol g 2.1 exhibits hydrophobic binding and has a high sequence identity (83.05%) with Sol i 2 from *S. invicta*. Notably, Sol g 2.1 acts as a mediator, causing paralysis in crickets. Given its structural resemblance and biological function, Sol g 2.1 may play a key role in transporting hydrophobic potent compounds, which induce paralysis by releasing the compounds through the insect’s nervous system. To investigate this further, we constructed and characterized the recombinant Sol g 2.1 protein (rSol g 2.1), identified with LC-MS/MS. Circular dichroism spectroscopy was performed to reveal the structural features of the rSol g 2.1 protein. Furthermore, after treating crickets with *S. geminata* venom, immunofluorescence and immunoblotting results revealed that the Sol g 2.1 protein primarily localizes to the neuronal cell membrane of the brain and thoracic ganglia, with distribution areas related to octopaminergic neuron cell patterns. Based on protein—protein interaction predictions, we found that the Sol g 2.1 protein can interact with octopamine receptors (OctRs) in neuronal cell membranes, potentially mediating Sol g 2.1’s localization within cricket central nervous systems. Here, we suggest that Sol g 2.1 may enhance paralysis in crickets by acting as carriers of active molecules and releasing them onto target cells through pH gradients. Future research should explore the binding properties of Sol g 2.1 with ligands, considering its potential as a transporter for active molecules targeting pest nervous systems, offering innovative pest control prospects.

## 1. Introduction

The tropical red fire ant (*Solenopsis geminata*) is widely distributed throughout Thailand. Its venom is composed of various compounds, especially four major small, soluble proteins called Sol g 1, 2, 3, and 4 [1]. Interestingly, Sol g 2 shares a significant sequence identity (83.05%) with Sol i 2 in *S. invicta*, both having five α-helices and three intramolecular disulfide bridges, forming a hydrophobic cavity. Sol i 2 is known for its strong binding affinity with hydrophobic molecules such as aphid alarm pheromone (*E*)-β-farnesene, ant alarm pheromone analogs, and hydrophobic compounds [2,3,4]. It is believed that Sol i 2 plays a crucial role in the transportation of fire ant hydrophobic compounds from the synthesis area to the reservoir sac, possibly forming complexes with some components in the venom duct before secretion. Notably, the structure of Sol i 2 closely resembles that of odorant-binding proteins (OBPs) found in the olfactory organs of various insect species [3,5].

OBPs are small, soluble proteins consisting of not over 150 amino acids and are part of the peripheral olfactory system. They act as mediators, which are located in sensillum lymph fluid, between odorant receptors (ORs) and odorants from the external environment. Additionally, ORs represent the most extensive subfamily of class A G protein-coupled receptors (GPCRs) [6,7]. Upon passing through sensilla pores, odorants bind to OBPs, and the bound odorants are then carried through the sensillum lymph to reach the sensory dendrites. This signal is transmitted to the antennal lobe or subesophageal ganglion of insects. Pheromone-binding proteins (PBPs) are part of the OBP family that specifically bind and transport molecules to pheromone receptors (PRs) [8,9]. Considering the high sequence identity and structural homology with Sol i 2, it has been hypothesized that Sol g 2 may also be involved in binding and transporting hydrophobic molecules such as piperidine alkaloids, which serve as nicotinic antagonists, resulting in rapid prey paralysis [10]. Moreover, previous studies have demonstrated that *S. geminata* crude venom, when injected into crickets, causes paralysis, and this effect can be reduced by the presence of the anti-Sol g 2 antibody in the venom. Additionally, a mixture of piperidine alkaloids and the Sol g 2 protein reduced paralysis in crickets [11]. Although Sol g 2 plays a crucial role in mediating the effects of the paralysis or numbing of crickets, it does not cause paralysis by itself [1,2,12]. Nevertheless, the precise mechanisms and locations of Sol g 2 are still unknown.

The behavior of insects is regulated by neurotransmitters, which collectively exert control over a wide range of vital biological processes necessary for the sustenance and functioning of insects [13,14,15]. Octopamine (OA) is one of the insect’s main neurotransmitters. It binds to specific superfamilial GPCRs, which are a class prominently situated within the central nervous system of diverse insect species, cricket populations included [15,16]. In the context of crickets and other insects, octopamine receptors (OctRs) assume a pivotal role in the modulation of diverse behavioral patterns. For instance, during periods marked by heightened stress, OA neurotransmitters can instigate flight responses or amplify locomotive activity. Furthermore, they wield influence over aggression and mating behaviors [17,18]. This may relate to the role of Sol g 2 in enhancing the paralysis factor in crickets [11,12]. Consequently, it is conceivable that the Sol g 2 protein, which is similar to the LmaPBP (a pheromone-binding protein from the cockroach *Leucophaea maderae*) may serve as a carrier, releasing its ligands when triggered by the pH gradient at the cellular membrane, as previously suggested [19,20,21]. This protein possibly carries hydrophobic molecules or potent molecules in the *S. geminata* venom and then binds to receptors on the neuronal cell membrane, leading to the obstruction of neurotransmitter signaling by competitive binding as well as toxicity from these molecules [22,23,24]. In this context, our primary objective was to investigate the specific localization of the Sol g 2.1 protein after being injected into crickets to predict the possible role of action in inducing paralysis in crickets.

Thus, in this study, we produced the recombinant Sol g 2.1 protein (GenBank: UYX46120.1), and it was then identified with an LC-MS/MS analysis. Next, circular dichroism spectroscopy (CD) was not only used to investigate Sol g 2.1 protein features, but it also was used to explore the protein conformation change associated with pH dependency. Importantly, we also studied the specific localization and the intact structure of the Sol g 2.1 protein after being injected using immunofluorescence and immunoblotting analyses, respectively. Moreover, protein—protein docking was utilized to predict the Sol g 2.1 protein and its possible receptor binding. These findings may contribute to understanding the potential role of Sol g 2.1 as a mediator in enhancing the activity of *S. geminata* venom. Furthermore, it may be developed as a potential strategy for innovative approaches to pest control in the future.

## 2. Results

### 2.1. Expression, Purification, and Identification of rSol g 2.1 Protein

The mature Sol g 2.1 protein (GenBank: UYX46120.1) was composed of 119 amino acids encoded from 360 bps, as confirmed with Edman degradation sequencing (Appendix A). Conversely, the mature sequence of the Sol g 2.1 protein, as computed with ExPAsy, was 13,386.68 Da (http://web.expasy.org/compute_pi/, accessed on 26 November 2022). After protein expression, an extensively purified fraction of rSol g 2.1 (approximately 17 kDa) was analyzed using LC-MS/MS for identification (Appendix A). Peptides from this fraction were then subjected to identification using the MASCOT search engine, which was employed in conjunction with the NCBI databases. Theoretical calculations using ExPAsy online software indicated that the molecular weight (MW) of the purified rSol g 2.1 protein band was approximately 15,966 Da. However, during experimental protein purification, the observed MW was found to be around 17 kDa (accounting for the 6X-Histidine tag, which added approximately 1 kDa). This deviation from the theoretical MW aligned with prior research findings, which also reported rSol g 2.1 to be approximately 16 kDa [11]. Such congruence in MW values strengthened the validity of our results. Furthermore, the amino acid sequence of Sol g 2.1 displayed a remarkable resemblance to venom allergen 2, an allergen protein found in *S. invicta* venom (accession no: XP_011156057.1), suggesting a potential allergenic role for rSol g 2.1 (Table 1).

### 2.2. Secondary Structure of Sol g 2.1

The secondary structure composition of the rSol g 2.1 protein was investigated using circular dichroism (CD) spectroscopy. The CD spectra of rSol g 2.1 protein displayed a reduction in intensity at 208 and 222 nm under pH 5.5 conditions compared with those at pH 7.3 (Figure 1A). Subsequently, the CD spectra were analyzed using the K2D3 online software to quantify the percentage of α-helix content. At pH 7.3, the secondary structure composition of the rSol g 2.1 protein consisted of 51.03% α-helices, 8.95% β-strands, and 40.02% unidentified structures. Conversely, at pH 5.5, the protein exhibited 39.37% α-helices, 11.52% β-strands, and 49.11% unidentified structures. Both measurements showed a maximum error of more than 0.32. These findings unequivocally indicated that the α-helix content of the rSol g 2.1 protein was significantly higher at pH 7.3 compared with pH 5.5. Furthermore, the protein’s near-UV wavelength was analyzed following its dissolution in pH 5.5 and 7.3 buffers. The results revealed a notable decrease in the intrinsic fluorescence of the rSol g 2.1 protein at lower pH values. This decrease in fluorescence suggested that aromatic residues in the protein’s inner cavity experienced increased exposure to the solvent under acidic conditions (Figure 1B). As a consequence, it appears that the rSol g 2.1 protein underwent a loss of its rigid tertiary structure when dissolved in pH 5.5 [25]. Furthermore, there was a plausible indication that this phenomenon may correspond to the liberation of its endogenous ligands in proximity to the cellular membrane [26].

### 2.3. Paralysis Symptoms and Aggressive Behavior

In the treatment with the rSol g 2.1 protein, no paralysis symptoms were observed in crickets, even at the highest concentration of the protein (0.850 µg/g body weight). In contrast, when injected with mixtures of the rSol g 2.1 protein and various concentrations of 2-methyl piperidine, the results showed a significant decrease in the PD_50_ to 0.098% (*v*/*v*) 2-methyl piperidine, compared with 2-methyl piperidine alone (PD_50_ = 0.116% (*v*/*v*)). Therefore, the mixture of the Sol g 2.1 protein and 2-methyl piperidine reduced the PD_50_ when compared with 2-methyl piperidine alone, as shown in Figure 2. These results indicated that the Sol g 2.1 protein acted as an important mediator in enhancing the paralytic activity of 2-methyl piperidine (solenopsins analog) as well as in *S. geminata* crude venom after being injected into crickets [11,12,27]. In crude venom treatment, the PD_50_ was approximately 66 µg/g body weight, which was set as a positive control (Appendix A).

### 2.4. Specific Localization of Sol g 2.1 Protein with Immunofluorescence Analysis

The crickets underwent an experimental procedure involving the administration of *S. geminata* crude venom with a PD_50_ concentration. Within the treatment group, the presence of the Sol g 2.1 protein was identified through immunolabeling, specifically localized within numerous cell bodies encompassing the dorsolateral region and protocerebrum (DL and Pr), as depicted in Figure 3H [28]. Evidently, the protocerebral cell bodies are marked with red arrowheads (Figure 3K). Additionally, the cells were discernible in the vicinity of the deutocerebrum (De) and the pars intercerebralis (PI), indicated by red arrowheads (Figure 3N) [29]. Moreover, distinct localization patterns of the Sol g 2.1 protein were detected, extending to the ventral cell surface of the brain, which were present in immunopositive cells, identifiable through the white arrowhead within Figure 3N [30,31]. In the thoracic ganglia section, Sol g 2.1 protein immunopositivity was evident along the cell’s surface at the ganglion’s ventral surface that were shown in the red and white boxes (Figure 4H). Furthermore, instances of the Sol g 2.1 protein were also discerned in connective cells linking various ganglia, indicated by a white arrowhead (Figure 4K). Conversely, in the control group, no discernible indication of the targeted protein immunoreactivity was identified (Figure 3C–E and Figure 4C–E). The negative control sections did not exhibit any immunoreactivity, as demonstrated in the Appendix A.

### 2.5. Immunoblotting

To confirm the specificity of binding between the Sol g 2.1 protein and anti-Sol g 2.1 protein antibodies, a purified rSol g 2.1 protein was analyzed using an immunoblotting analysis. The results revealed that immunoreactivity appeared at approximately 17 kDa, which was the rSol g 2.1 protein’s expected size (Figure 5A). Therefore, the primary antibody had high specificity with the Sol g 2.1 protein. Furthermore, we found that the localization of the Sol g 2.1 protein after being injected into the crickets appeared in the cell membrane of the brain and thorax tissues (Figure 3 and Figure 4). Importantly, to determine the intact protein structure after penetrating, allowing the hemolymph flow of the crickets, we also performed immunoblotting. The results showed the immunoreactivity of the Sol g 2.1 protein (~15 kDa), the samples of which were extracted from the treated compared to non-treated crickets (Figure 5B and Appendix A). Additionally, there was a more intense band of Sol g 2.1 protein in the head than in the thorax of the crickets’ extraction samples. Therefore, we believed that the Sol g 2.1 protein could penetrate following the hemolymph of the cricket and specifically be located on the cell membrane of the cricket’s brain and thorax with intact forms.

### 2.6. The Overall Structure and Structural Homology of Sol g 2.1 Protein

The mature deduced sequence of the Sol g 2.1 protein consisted of 119 amino acids (GenBank: UYX46120.1). A comparative analysis of Sol i 2 (*S. invicta*) and Sol r 2 (*S. richteri*) revealed sequence identities of 83.05% and 86.32%, respectively (Appendix A). To model the three-dimensional structure of Sol g 2.1, we utilized the Swiss-Model program with the crystallized structure of Sol i 2 (PDB ID: 2ygu.1.A, resolution of 2.60 Å) as a template. The resulting three-dimensional structure of Sol g 2.1 showed a protein organization featuring five α-helices arranged around a central hydrophobic cavity. The stability of this structure was reinforced by three intramolecular disulfide bridges (Cys15–Cys38, Cys62–Cys75, and Cys82–Cys103) and one intermolecular disulfide bridge with an identical monomer at Cys22. The helices at positions α2–α5 form a hydrophobic pocket, creating a potential binding site for hydrophobic molecules (Figure 6A). Furthermore, we assessed the homology structure of Sol g 2.1 using a Ramachandran plot, with amino acids represented by green and brown crosses corresponding to the Phi (*X*-axis) and Psi (*Y*-axis) angles. The plot indicated that 98.261% of the amino acids fell within the highly preferred zone, and 1.739% within the acceptable zone (Figure 6B). This suggested that the 3D model of the Sol g 2.1 protein was of acceptable and good quality [35,36]. Additionally, residues within the α-helical and β-strand structures were situated in the bottom and top-right quadrants of the plot, respectively. The predicted three-dimensional structure offered valuable insights into the potential binding site of Sol g 2.1 and its capability to interact with hydrophobic molecules, providing further support for its proposed role as a transporter protein. This high-quality 3D model lays the foundation for future investigations into the functional aspects of Sol g 2.1 and its potential applications in pest control strategies.

Despite the relatively low sequence identity of 15.49% between Sol g 2.1 and the LmaPBP (PDB ID: 1org.1.A, resolution of 1.7 Å), both proteins exhibited four similar helical folds, as demonstrated in Figure 6C. The LmaPBP’s structure comprised six helices and was stabilized by three disulfide bridges formed by Cys18–Cys49, Cys45–Cys106, and Cys94–Cys115. On the other hand, Sol g 2.1 contained three disulfide bridges at the positions Cys15–Cys38, Cys62–Cys75, and Cys82–Cys103. Utilizing the MOE software to superimpose the positions of Sol g 2.1 (α2-α5) onto the LmaPBP (α3–α6), the root-mean-square deviation (RMSD) between the two proteins was calculated to be 0.55 Å, as depicted in Figure 6D. It is worth noting that the RMSD values in the range of 1.5 to 2 Å are considered acceptable in structural alignments [37]. The hydrophobic cavity of the LmaPBP was confined by the α3–α6 positions, while the corresponding cavity in Sol g 2.1 was surrounded by the 2α–5α positions. Notably, the structure of the LmaPBP featured an open pocket that was readily accessible to the bulk solvent [38].

### 2.7. Sol g 2.1 Protein—Octopamine Receptor Docking

Our investigation entailed protein—protein docking simulations involving the Sol g 2.1 protein—OctRs complex to elucidate the pivotal residues participating in the interaction. The protein—protein docking procedures were executed using the ClusPro 2.0 server, yielding five distinct clusters of docked complexes. The preeminent cluster, encompassing 113 members, exhibited the most favorable outcome with the lowest energy of −1230.9. Accordingly, this top cluster was meticulously scrutinized to unravel the intricate interplay between the receptor and ligand, thereby enabling the identification of critical residues responsible for the interaction. Furthermore, the docking outcomes revealed the involvement of specific residues of Sol g 2.1 protein—OctR, namely E111-R190 and R183, D28-R180, R106-Y178, L105-W179, R94-Y168, R91-F470, and S92-P474 in the formation of hydrogen bond interactions. Figure 7 illustrates the binding interaction of the protein–receptor complex using PyMol.

As mentioned, OctRs are located at the neuronal cell membrane and then transduce the signaling pathway, leading to specific physiological responses and behaviors of insects [39]. Hence, one plausible mechanism for the localization of the Sol g 2.1 protein on the cell membrane within the cricket’s central nervous system could involve the binding of the Sol g 2.1 protein to OctRs (Figure 7). Exogenous ligands engage with the receptor within the invertebrate nervous system. This interaction leads to heightened nervous activity, resulting in paralysis and delivering a wide-ranging insecticidal effect [40].

## 3. Discussion

The Sol g 2.1 protein holds a prominent position as a major allergenic constituent within *Solenopsis* venom, a category recognized for its venom allergen proteins renowned for their heightened immunogenicity. Beyond its established role in triggering allergic responses, emerging evidence suggests that Sol g 2.1 may also wield a significant influence in the induction of paralysis in insects. Prior research has revealed that Sol g 2 has the capacity to enhance the paralytic effects of piperidine alkaloids, which are the major potential molecules in the crude venom of *S. geminata*, when injected into crickets [11,12]. This intriguing finding suggests that Sol g 2 likely functions as a pivotal protein responsible for the binding and transportation of hydrophobic molecules from the injection site to critical sites within the nervous system of insects, ultimately culminating in paralysis. However, the precise mechanisms governing how Sol g 2 enhances the paralytic effect in insects are yet to be fully elucidated.

In the present study, we embarked on a comprehensive characterization of the secondary and tertiary structures of the rSol g 2.1 protein while delving into its specific localization within the context of its paralysis-enhancing role. Verification of the rSol g 2.1 protein construct was identified through an LC-MS/MS analysis, which unveiled its high similarity with venom allergen 2 from *S. invicta* [2,11]. Circular dichroism spectroscopy was employed to assess the secondary structure of rSol g 2.1, revealing a higher proportion of α-helix content in comparison with those of β-turn and random coils. Notably, rSol g 2.1 displayed optimal stability in a pH 7.3 buffer, but under pH 5.5 conditions, it underwent conformational alterations, resulting in decreased α-helix content. This shift was attributed to the exposure of aromatic amino acids to the solvent at lower pH, thereby transitioning the protein from closed to open cavity binding, ultimately leading to the loss of its rigid tertiary structure, as indicated by the near-UV CD spectra [25]. From these findings, we believe that the Sol g 2.1 protein possibly binds and carries the potential molecules in *S. geminata* venom from the site of injection, and then releases them through the targeted cell membrane, allowing for a different pH environment [41]. Additionally, the shifting of the binding pocket after becoming exposed to the solvent was responsible for the release of the ligand at a pH level of 5.5 [19,26,42,43].

In terms of its role in paralytic activity, our study revealed that the Sol g 2.1 protein enhanced the paralytic effects of 2-methyl piperidine and solenopsin analogs [44,45]. It is important to note, however, that the Sol g 2.1 protein itself did not induce paralysis in crickets, consistent with previous research findings [11,12]. As suggested, the Sol g 2.1 protein may function as a hydrophobic carrier protein, potentially shielding the hydrophobic portion of piperidine alkaloid ligands and subsequently releasing them at specific receptors on the cell’s surface [2,3]. Piperidine derivatives disrupt the coupling process of the binding of acetylcholine to receptors and act as nicotinic antagonists. Subsequently, an increase in ionic conductance leads to the depolarization of the target cell and the manifestation of symptoms. Fast depolarization blockades also result in paralysis [46,47,48,49]. As we know, the Sol g 2.1 protein plays a crucial role in cricket paralysis, likely related to the insect’s central nervous system. Nevertheless, the specific location of the protein has not been revealed. Herein is our exploration of Sol g 2.1’s specific localization. We administered the protein solution into the abdominal cavity of crickets, revealing its localization predominantly at the membrane of cell bodies encompassing critical regions, including the dorsolateral aspect of the protocerebrum, pars intercerebralis, protocerebrum, and the central body. Sol g 2.1 protein immunopositivity was also presented at the cell’s surface at the ganglion’s ventral surface and connective cells between each ganglion. The distribution areas of the Sol g 2.1 protein in the cricket’s central nervous system are close to the octopaminergic system [13,50,51,52]. OctRs, vital components located within the neuronal cell membrane, serve as mediators in transducing signaling pathways that govern specific physiological responses and behaviors in insects [39]. Consequently, one plausible mechanism accounting for the localization of the Sol g 2.1 protein on the neuronal cell membrane within the cricket’s central nervous system could entail its interaction with OctRs, predicted with protein—protein docking [53]. In short, the Sol g 2.1 protein may transfer potent ligands and release them through OctRs in the nervous system, leading to heightened nervous activity, paralysis, and a broad insecticidal effect.

In insects, proteins and other fluids are distributed allowing hemolymph flow. They are distributed from the abdomen cavity and pass through the dorsal vessel. Hemolymph and proteins are pumped from the hind end passing to a valved chamber series (ostia) and aorta. The hemolymph fluids are moved forward by accessory pumps through all organs of the insect before moved back again to the abdomen [54]. According to this study, to precisely determine the localization of the Sol g 2.1 protein, we administered *S. geminata* venom into the crickets’ abdomen. After being injected, the entire of venom constituents underwent systemic dispersion via the insect’s hemolymph flow. Notably, in the venom components, phospholipases (Sol g 1) among others promote diffusion, and these proteins have the capacity to induce tissue lesions, thereby facilitating the diffusion of venom [1,55]. This phenomenon enables the Sol g 2.1 protein to readily traverse tissue barriers and subsequently bind to OctRs located on the cell membrane of the CNS within the cricket (Figure 8). However, to identify the particular GPCRs that interact with the Sol g 2.1 protein, it is necessary to conduct additional research into alterations at the second messenger level, such as Ca^2+^ or cAMP [56].

In forthcoming investigations, our objective will be to explore the binding properties of the Sol g 2.1 protein with various ligands, further probing its potential as a transporter protein capable of conveying active hydrophobic molecules, such as solenopsin or solenopsin ingredients, with a targeted delivery to the nervous systems of pests. Such research endeavors hold the promise of opening innovative pathways into the development of pest control strategies, potentially revolutionizing how we address pest management in the future.

## 4. Materials and Methods

### 4.1. Expression and Purification of Recombinant Sol g 2.1

Sol g 2.1-pProEx-HTB recombinant plasmid was provided by Dr. Hathairat Buraphaka [57]. After DNA sequencing, the plasmids were transformed into *E. coli* BL21 (DE3)-competent cells, following the established protocol [12]. For large-scale culturing, a starter culture was inoculated into LB medium and incubated at 37 °C until the cell optical density reached 0.6 (OD_600_). Subsequently, the cell culture was induced by adding 0.4 mM IPTG at 37 °C. After an incubation period of eight hours, the induced culture was harvested and suspended in a lysis buffer. The cells were subjected to three cycles of freeze—thaw to facilitate cell lysis, followed by sonication for further disruption. The insoluble fraction was then denatured by stirring at 4 °C for 6 h in an 8 M urea solution, 1 mM DTT, and 0.1 mM PMSF in a 50 mM Tris-HCl buffer at pH 7.4. To refold the denatured protein, urea was gradually removed with dialysis at 4 °C. The refolded rSol g 2.1 protein was purified using ion-exchange chromatography with a gradient elution process on the AKTAprime plus system (GE Healthcare). The binding buffer for the equilibrium step contained 20 mM Tris-HCl at pH 7.4, 500 mM NaCl, and 20 mM imidazole. The elution buffer consisted of 20 mM Tris-HCl at pH 7.4, 500 mM NaCl, and 500 mM imidazole. Each eluted fraction was observed using 13% SDS-PAGE, and only fractions without nonspecific proteins were pooled and dialyzed to remove imidazole. To confirm the rSol g 2.1 protein, it underwent LC-MS/MS analysis, allowing for the accurate characterization of the protein.

### 4.2. Identification of rSol g 2.1 Protein Using LC-MS/MS

The protein band of interest, approximately 17 kDa in size, was carefully excised from the gel and subjected to LC-MS/MS analysis for identification. The excised band was washed to remove any contaminants and then digested using modified trypsin (Promega, USA) at a concentration of 20 ng/spot in a solution containing 50% acetonitrile and 10 mM ammonium bicarbonate. The digestion process took place at 37 °C and lasted for three hours. Following digestion, the supernatant containing the digested peptides was collected and dissolved in a solution of 0.1% (*v*/*v*) formic acid. The sample was then dried at 37 °C for three hours to concentrate the peptides. The resulting material was prepared for analysis using the Ultimate 3000 LC System (Dionex) coupled with an ESI-Ion trap MS (HCTultra PTM Discovery System, Bruker Daltonik). In the search for peptide identification, a local MASCOT server was utilized, and the SwissProt and NCBI protein databases were referenced. The peptides were considered to have potentially missed cleavages due to trypsin digestion. Methionine oxidation was considered as a variable modification, while carbamidomethyl was considered a fixed modification. Monoisotopic mass data were utilized for the search process [11,12].

### 4.3. Characterization of rSol g 2.1 Protein

To analyze the secondary and tertiary structures of the rSol g 2.1 protein, analysis of the protein was conducted at the Faculty of Science, Khon Kaen University, using circular dichroism (CD) spectra. The CD spectra were recorded with a bandwidth of 5 nm and a scan speed of 50 nm/min. Far-UV measurements were performed using a 1 mm cuvette made of quartz, covering a wavelength range of 190–260 nm, while near-UV measurements were taken at wavelengths ranging from 250 to 320 nm. To prepare the samples for analysis, the rSol g 2.1 protein was dissolved into two different buffers: 20 mM ammonium acetate at pH 7.3 and 20 mM sodium acetate at pH 5.5. The final concentration of the protein in both buffers was adjusted to 0.2 mg/mL, following a previously established protocol [37,58]. To determine the content of α-helices in the rSol g 2.1 protein, the obtained CD spectra were subjected to analysis using the K2D3 online software (http://cbdm-01.zdv.uni-mainz.de/~andrade/k2d3/, assessed on 28 January 2021). This allowed for the characterization and quantification of the protein’s secondary structure, particularly the α-helical content [36].

### 4.4. Paralysis Activity

Adult male crickets (*Acheta domestica*) were acquired from the NL Cricket Farm in Khon Kaen, Thailand, and were identified by a veterinarian prior to use in this study [14,59]. The crickets were placed in a plastic box with open-air conditions at 30 °C and a relative humidity of approximately 50–60% for 24 h (12 h of light, 12 h of dark). Crickets with an average body weight of 0.4 g were selected. Prior to testing, a range of doses of 2-methyl piperidine (0.1, 0.4, 0.3, 0.2, and 0.1% (*v*/*v*) in PBS) and crude venom (200, 150, 100, and 50 µg/g body weight (BW)) were initially administered to the crickets. Under our observation for more than 1 h, we found that 0.3% (*v*/*v*) 2-methyl piperidine and 100 µg/g BW of *S. geminata* venom were minimum dosages to elicit a 100% paralysis observed in the crickets. These specific dosage levels were subsequently designated as the initial doses for the experiment [60,61]. For the paralytic dose (PD_50_) examination, we observed three treatment groups with various concentrations of rSol g 2.1 (0.85, 0.43, 0.21, and 0.11 µg/g BW), 2-methyl piperidine in PBS (0.3, 0.2, 0.1, 0.05, 0.025, and 0.012% (*v*/*v*)), and mixtures of the highest dose of rSol g 2.1 (0.85 µg/g BW) and each concentration of the piperidine solution. For the positive treatment, *S. geminata* crude venom was collected through their stingers by meticulously gathering it drop-by-drop using a capillary tube, then it was dissolved into a pH 7.4 PBS buffer to an approximate total protein concentration of 4 µg/µL (200 µg/g BW, stock), measured with Bradford’s method. Next, the crude venom stock was dissolved into PBS to final concentrations of 100, 75, 50, and 25 µg/g BW. PBS buffer was used as a non-treatment group. In this procedure, 20 µL of individual treatment was injected into the abdominal cavity of crickets (six crickets in triplicate), following the previously described protocol [11,12,27,62]. After 30 min, crickets that could not overturn from the dorsal position and had no locomotion were considered to be paralyzed. The PD_50_ value of each treatment group was compared with an unpaired *t*-test analysis using GraphPad Prism 9 (GraphPad Software, San Diego, CA, USA). The group of crickets that received *S. geminata* crude venom treatment (PD_50_) was selected for further investigation of the protein’s localization using immunofluorescence and immunoblotting techniques.

### 4.5. Specific Localization of Sol g 2.1 Protein with Immunofluorescence

Following the paralysis assay, the crickets from two treatment groups were promptly dissected into two parts, including the head and thorax-abdomen. All dissected parts were fixed in 10% neutral formalin at 4 °C for 18 h. Subsequently, they were washed multiple times with 70% ethanol by gently shaking, followed by additional washing with chilled PBS at pH 7.4. The dissected parts were then immersed in 10% EDTA solution for 48 h, with the EDTA solution changed every 12 h during this period. After the 48-h EDTA treatment, the dissected tissues were washed again with chilled PBS at pH 7.4 and then soaked in 70% ethanol for dehydration. The tissues were subjected to standard histological processing, which involved dehydration using an alcohol series (70–100%), followed by clearing with xylene, infiltration, and embedding in paraffin wax. Subsequently, 6 µm thick tissue sections were cut using a microtome (HESTION, ERM 4000), placed on coated slides, and dried at a warm temperature for 12–16 h. To prepare the tissue sections for immunohistochemistry, the slides were deparaffinized with xylene and rehydrated through a series of graded ethanols (100–70%). Endogenous peroxidase was blocked by shaking the tissues in 3% hydrogen peroxide in methanol (H_2_O_2_). The tissues were then immersed in 0.05% Tween-20 in PBS at pH 7.4 for 5 min and subsequently washed with PBS, 1% glycine, and PBST (PBS with Tween-20) to ensure proper washing. To block non-specific binding, the tissues were incubated with 4% BSA in PBST for 2 h before applying the primary antibody. The primary antibody used was a rabbit anti-Sol g 2.1 protein polyclonal antibody (sequence: CVDRETQRPRSNRQ, Lot: U743UIA090-1) with 1:200 dilution in the blocking solution, and it was applied to all tissues at 4 °C for 18 h. In the negative control sections, the pre-immune serum was applied instead of the rabbit anti-Sol g 2.1 protein polyclonal antibody. Following the primary antibody incubation, the tissues were washed multiple times with PBS and PBST. After washing the primary antibody, the secondary goat anti-rabbit IgG Alexa Fluor^®^ 488 (Abcam, USA, Lot: 1616933) at a dilution of 1:1000 was applied to the tissues and incubated at 4 °C for 2 h. DAPI (Invitrogen™, USA) was used as a nuclear staining. Following the mounting process, the tissues were examined using confocal microscopy (Olympus FV1000, Tokyo, Japan) for immunofluorescence detection [63,64].

### 4.6. Immunoblotting

To determine the intact and specific Sol g 2.1 protein in the crickets after being injected, we performed an immunoblotting analysis. The crickets that were treated with the highest concentration of rSol g 2.1 protein solution without paralysis symptoms were selected. In this procedure, 15 crickets were immediately frozen at −20 °C, and they were then dissected into two segments (head and thorax). After each tissue segment was ground under liquid nitrogen, they were then homogenized in lysis buffer (0.5% Triton-X 100, PMSF, 20 mM Tris-HCl pH 7.4, 10 mM EDTA, 12 mM NaCl, and 50 mM KCl,). After that, the homogenated soluble fraction was separated from others with centrifugation. The supernatant was carried out to lyophilize and then dissolved into 20 mM Tris-HCl to a final concentration of 10 mg/mL [65]. Afterward, the samples were separated into 13% SDS-PAGE. These resolved proteins were subsequently transferred onto nitrocellulose blots. To prevent non-specific binding, the blots underwent a blocking procedure using a solution containing 5% skim milk within Tris-buffered saline supplemented with 0.1% Tween-20 (TBST). Following the blocking step, the blots were subjected to overnight incubation with rabbit anti-Sol g 2.1 protein primary antibodies (1:500 in blocking solution) at 4 °C. Subsequent to the primary antibody incubation, thorough washing with TBST was conducted. This was followed by incubation with the secondary antibodies (goat anti-rabbit IgG (H + L)-linked HRP, 1:2000 in blocking solution) for 2 h at room temperature. After an additional round of TBST washing, the visualization of protein bands was accomplished utilizing a DAB Advanced Chromogenic Kit (eBioscience™, Invitrogen, Waltham, MA, USA).

### 4.7. Homology Modeling

The known amino acid sequence of the Sol g 2.1 protein (Sol g 2.1, GenBank: UYX46120.1) was identified from a protein sequencing experiment. Next, the amino acid sequence was analyzed using the NCBI database (http://www.ncbi.nlm.nih.gov/, accessed on 6 November 2022), and Clustal Omega (https://www.ebi.ac.uk/Tools/msa/clustalo/, accessed on 24 November 2022) was used for the alignment of the sequence identity. The molecular weight and isoelectric points were obtained using ExPASy Bioinformatics (https://www.expasy.org/, accessed on 26 November 2022). Then, a homology model was generated based on a template by using the Swiss-Model program (https://swissmodel.expasy.org/, accessed on 10 December 2022), and we plotted torsion angles for estimating quality with online software for Ramachandran plots (https://zlab.umassmed.edu/bu/rama/, accessed on 1 December 2022) [66]. The sequence of Sol g 2.1 is similar to that of the LmaPBP, which was chosen as a template for the superposition of homology models. The root-mean-square deviation (RMSD) was evaluated for the comparison of the models, as determined with the Molecular Operating Environment (MOE) software 2019.01 (Chemical Computing Group Inc., 2011, Montreal, Canada).

### 4.8. Protein—Protein Docking

For our analytical endeavors, we retrieved crystallographic structures from the Protein Data Bank (PDB). Specifically, we employed the three-dimensional structure of Sol g 2.1 with the Swiss-Model program, utilizing the crystallized structure of Sol i 2 (PDB ID: 2ygu.1.A, 2.60 Å) as a template. The protein was complexed with octopamine receptor 1 (OctRs), partial (*Gryllus bimaculatus,* GenBank: BAV93772.1). The protein—protein docking was analyzed on ClusPro 2.0, a reputable protein—protein docking server (https://cluspro.bu.edu/login.php, accessed on 10 August 2023) [67]. ClusPro is renowned within the scientific community for its proficiency in conducting protein—protein docking tasks, as evidenced by its successful performance in the Critical Assessment of Prediction of Interactions (CAPRI) competition [68]. The methodology behind ClusPro hinges on a correlation approach known as PIPER [69]. PIPER is responsible for computing the energy associated with docked conformations within a grid, utilizing the efficient Fast Fourier Transform (FFT) technique in conjunction with pairwise interaction potentials. This advanced technique, powered by the precision of pairwise interaction potentials within PIPER, resulted in the retention of a notably reduced number of near-native structural configurations. Subsequently, the generated structural configurations underwent a clustering process, which employed pairwise root-mean-square deviation (RMSD) as the primary distance metric [70]. This clustering was carried out to optimize the representation and organization of the resulting structures. After that, the best-posed model was visualized with the PyMol program.

## 5. Conclusions

The Sol g 2.1 protein is a major allergenic component in *Solenopsis* venom, recognized for its strong immunogenic properties. Emerging evidence suggests that Sol g 2 may also significantly contribute to inducing paralysis in insects. In this study, a comprehensive analysis of the secondary and tertiary structures of the rSol g 2.1 protein was investigated after protein expression. Circular dichroism spectroscopy revealed a predominance of α-helix content in rSol g 2.1, with pH-dependent stability, shifting from closed to open cavity binding at lower pH levels. This suggested that Sol g 2.1 may act as a carrier for hydrophobic molecules in *S. geminata* venom, releasing them in response to varying pH conditions. Our exploration of the Sol g 2.1 protein’s specific localization concerned its paralysis-enhancing function. Sol g 2.1 did not induce paralysis by itself, but our study demonstrated its ability to enhance the paralytic effects of 2-methyl piperidine. Possibly, Sol g 2.1 functions as a hydrophobic carrier protein, and releases piperidine alkaloid ligands at specific cell surface receptors. Furthermore, the investigation into Sol g 2.1’s localization in crickets revealed its presence mainly on cell membranes within critical areas of the central nervous system, closely related to the octopaminergic system. This suggested a potential interaction between Sol g 2.1 and OctRs on neuronal cell membranes, which could explain its role in increasing nervous activity and inducing paralysis in insects. Future research should aim to explore Sol g 2.1’s binding properties with various ligands, further investigating its potential as a transporter protein for delivering hydrophobic active molecules to pests’ nervous systems. These endeavors hold the promise for innovative pest control strategies that could revolutionize pest management. This research not only deepened our understanding of insect—parasite interactions but also had practical applications with significant implications for the fields of entomology and neurobiology.

## Figures and Tables

**Figure 1 ijms-24-14814-f001:**
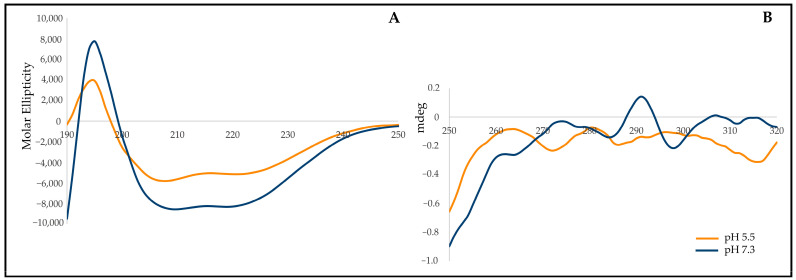
Circular dichroism spectra (CD spectra) of Sol g 2.1 protein in pH 5.5 (orange line) and 7.3 (dark blue line). (**A**) Far-UV. (**B**) Near-UV CD spectra.

**Figure 2 ijms-24-14814-f002:**
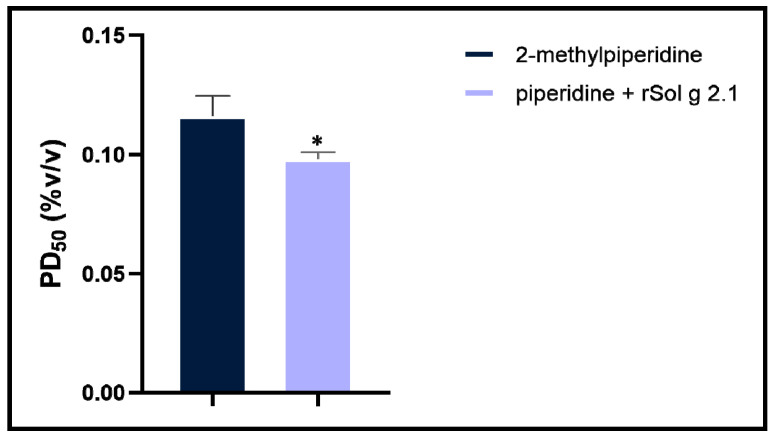
Column shows PD_50_ values (mean with SD) of 2-methyl piperidine and the mixture of rSol g 2.1 and various concentrations of 2-methyl piperidine groups after abdomenal injection into the crickets. (*) indicates significance (*p* < 0.05).

**Figure 3 ijms-24-14814-f003:**
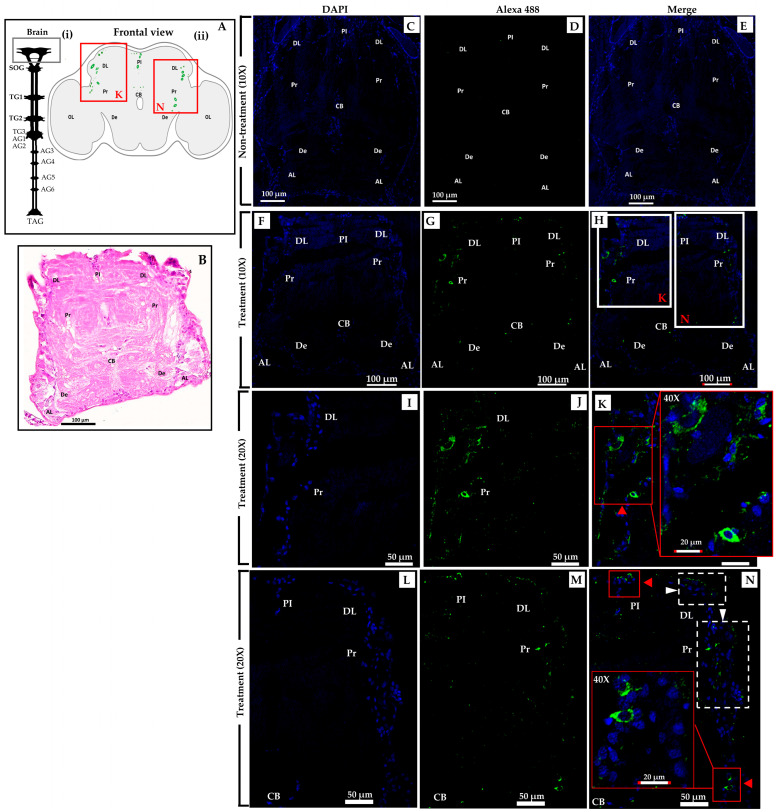
Immunofluorescent Sol g 2.1 detection photographs of the frontal sections of a cricket’s brain (*Acheta domestica*) after being treated with *S. geminata* venom. (**A**) Schematic diagram of the cricket central nervous system. (i) The gray box represents the frontal section of the brain. (ii) Schematic diagram of the distribution and distribution areas of the Sol g 2.1 protein in the cricket’s brain are represented by green-colored dots and red boxes, respectively. (**B**) Frontal section of cricket’s brain stained with hematoxylin and eosin (H & E). (**C**–**N**) Co-localization of DAPI nucleus staining (blue) and detection of the Sol g 2.1 protein with Alexa 488 (green) in the brain’s frontal sections. (**C**–**E**) The non-treatment group exhibited no detectable immunoreactivity. (**F**–**N**) The detection of the Sol g 2.1 protein and the nucleus in the brain’s frontal section tissue of the treated group. White boxes of (**H**) are enlarged by a scale of (20×) and shown in (**K**,**N**). The red arrowhead represents the immunoreactive detection of the Sol g 2.1 protein in the membranes of cell bodies. White arrowhead shows immunopositive cells. SOG = subesophageal ganglion; TG = thoracic ganglia (TG1–3); AG = abdominal ganglia (AG1–6); TAG = terminal abdominal ganglion (TAG); De = deutocerebrum; DL = dorsolateral region of protocerebrum; PI = pars intercerebralis; Pr = protocerebrum; AL = antenna lobe; CB = central body.

**Figure 4 ijms-24-14814-f004:**
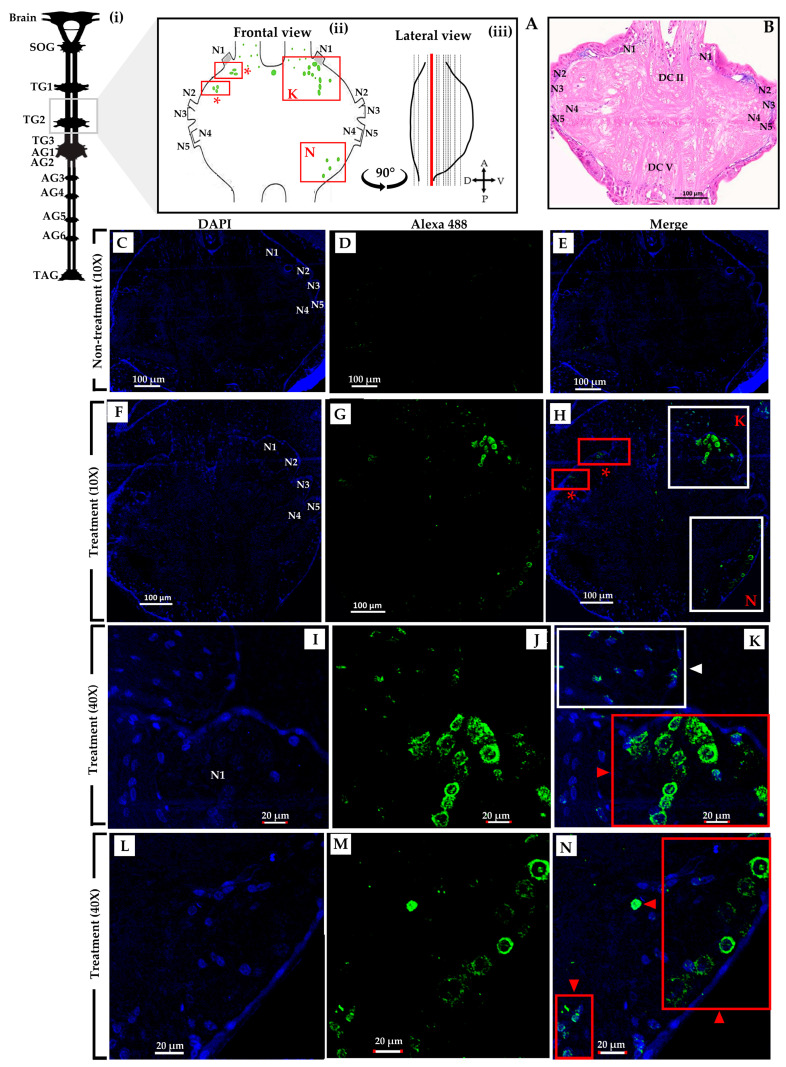
Confocal microscopic photographs of TG2 longitudinal section of a cricket (*Acheta domestica*) after being treated with *S. geminata* venom. (**A**) Schematic diagram of cricket central nervous system. (i) Gray box represents the TG2 longitudinal section. (ii) Schematic diagram of the distribution and distribution areas of the Sol g 2.1 protein in the TG2, which are represented by green-colored dots and red boxes, respectively. (iii) The schema of the lateral veiw of the TG2. Red lines indicate the site of the longitudinal section. (**B**) TG2 section, which was stained with H & E staining. (**C**–**N**) Co-localization of nucleus staining (blue) and the Sol g 2.1 protein shown in blue (DAPI) and green (Alexa 488), respectively. (**C**–**E**) The non-treatment group exhibited no detectable immunoreactivity. (**F**–**N**) Immunolabeling detection areas of the Sol g 2.1 protein in the thoracic ganglia tissue after being treated with *S. geminata* venom. White boxes of (**H**) represent the localization of the Sol g 2.1 protein and DAPI with a higher magnification scale (40×), which are shown in (**K**,**N**). The immunoreactivity of the Sol g 2.1 protein in the neuronal cell membrane is indicated with red arrowheads. White arrowhead shows immunopositive cells. SOG = subesophageal ganglion; TG = thoracic ganglia (TG1–3); AG = abdominal ganglia (AG1–6); TAG = terminal abdominal ganglion (TAG); N = nerve root (N1–5); DC = dorsal commissures (DCI–VI) [32,33,34].

**Figure 5 ijms-24-14814-f005:**
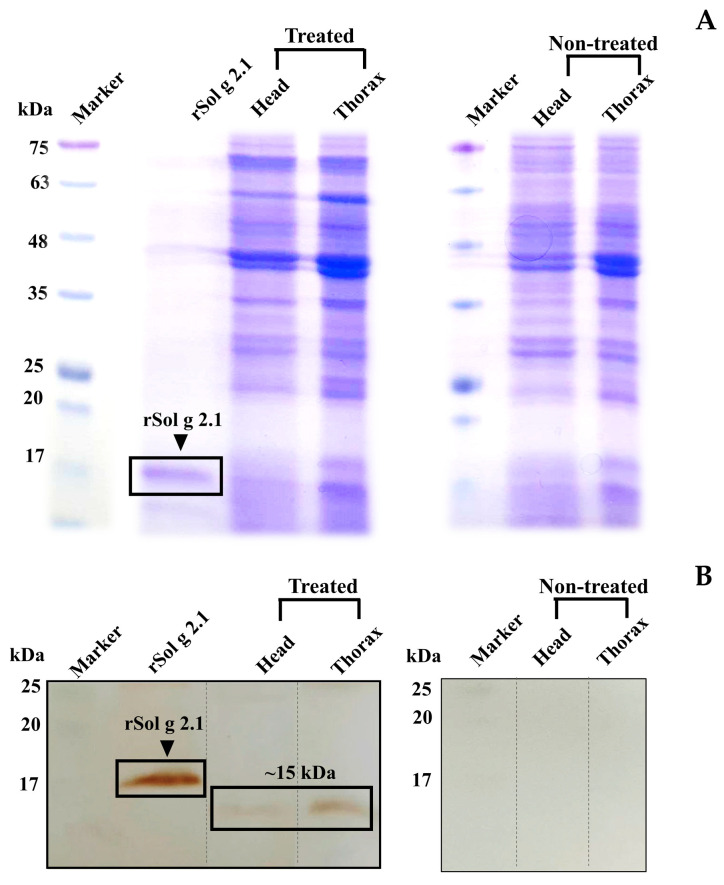
Comparative SDS-PAGE analysis (**A**,**B**) immunoblotting of the rSol g 2.1 protein and protein extraction from crickets (head and thorax) after being treated with a rSol g 2.1 protein solution.

**Figure 6 ijms-24-14814-f006:**
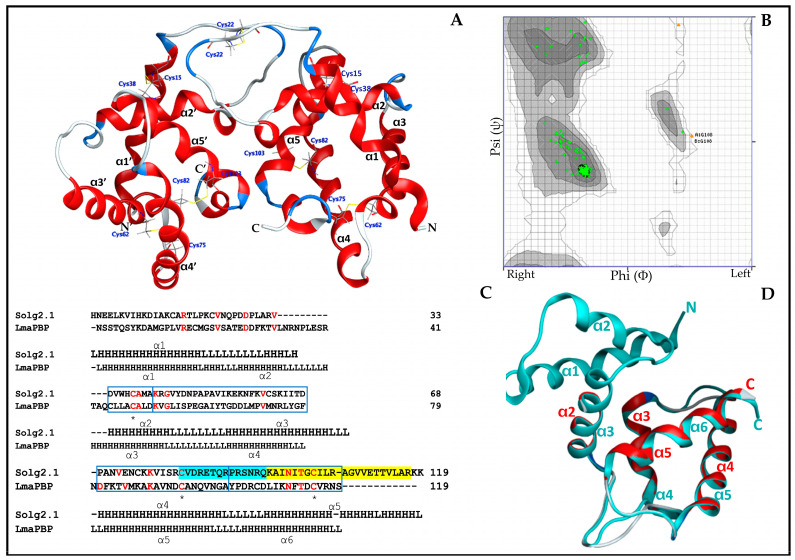
(**A**) Three-dimensional structure of the Sol g 2.1 protein. (**B**) Ramachandran plot: black, dark gray, gray, and light gray are the highly preferred zones. White with black grid is an acceptable zone. Green and brown crosses represent amino acid residues (generated by Ramachandran Plot Server76; https://zlab.umassmed.edu/bu/rama/, accessed on 12 November 2022). (**C**) Alignment of the deduced amino acid sequence of Sol g 2.1 and the LmaPBP. Red residues and asterisks (*) show amino acid identity and cysteine conserved residue, respectively. The amino acid sequence of Sol g 2.1 superimposed on the LmaPBP is shown in the blue box. Yellow shaded area represents the partial amino acid sequence that was verified by LC-MS/MS. The blue shade shows an amino acid sequence antigen in anti-Sol g 2.1 protein antibodies. (**D**) Superposition structure of the LmaPBP (turquoise; PDB ID: 1org.1.A) and Sol g 2.1 (red; PDB ID: 2gte).

**Figure 7 ijms-24-14814-f007:**
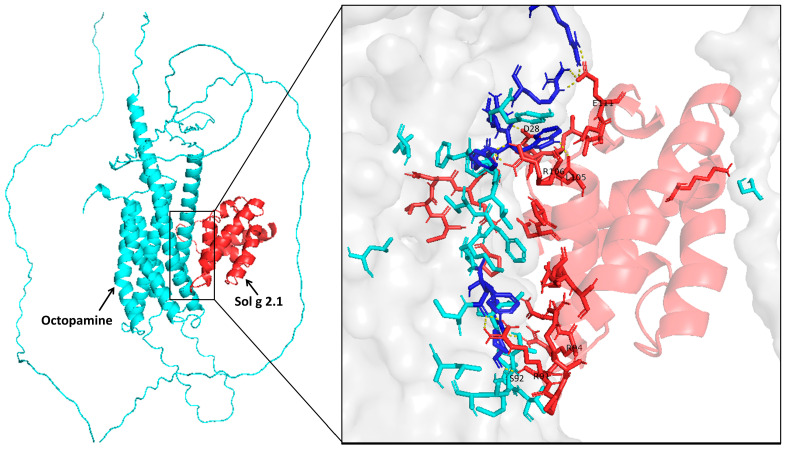
Docking of PyMol visualization of the GPCR receptor (OctRs, GenBank: BAV93772.1) and Sol g 2.1 protein complex with the most balanced set of scoring function that coefficients pose using the ClusPro 2.0 web server. The right panel shows insight into the OctR and Sol g 2.1 protein residue contact, which are represented by deep blue and red residues, respectively. Yellow dots show the interactions involved in protein receptor binding, whereas the gray surface is the receptor area.

**Figure 8 ijms-24-14814-f008:**
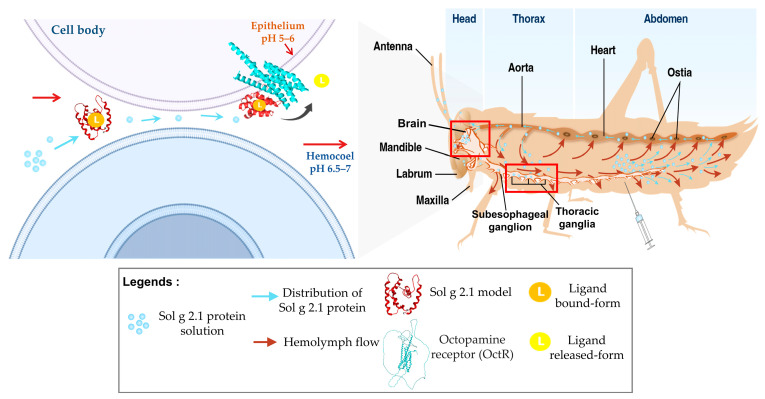
The illustration summary of hemolymph flow in crickets and the distribution of the Sol g 2.1 protein. Red boxes show the immunoreactive areas of Sol g 2.1 protein, after being injected *S. geminata* venom into cricket (**right**). Prediction of the function of Sol g 2.1 protein’s transportation and ligand release roles (**left**).

**Table 1 ijms-24-14814-t001:** Sol g 2.1 protein identification with LC-MS/MS.

Band	Theoretical MW ^1^	MW ^2^	Experiment MW ^3^	Matched Peptide Sequences ^4^	Score XC ^5^	Homologous Molecule
Purified band	15,966 Da	13,386.68 Da	17 kDa	R.AGVVETTVLAR.E K.AINITGCILR.A	155	Venom allergen 2; *S. invicta*

^1^ The theoretical molecular weight (MW) (average mass) was calculated using Mascot search results and LC-MS/MS. ^2^ The mature protein was calculated using the online software ExPASy Peptide Mass program (http://web.expasy.org/compute_pi/, accessed on 26 November 2022). ^3^ Experimental molecular weight. ^4^ Matched peptide sequence. All were obtained after LC-MS/MS analysis. ^5^ Score XC is the protein score obtained after LC-MS/MS analysis. A high score means good matching.

## Data Availability

The data of this study will be available upon request from the reader.

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
