# Peer review of "Characterization and Localization of Sol g 2.1 Protein from Solenopsis geminata Fire Ant Venom in the Central Nervous System of Injected Crickets (Acheta domestica)"

_ijms, 2023, doi:10.3390/ijms241914814_

Round 1

Reviewer 1 Report

Reviewer Comments #: -

The manuscript describes “Characterization of recombinant Sol g 2.1 protein from Solenopsis geminata fire ant venom and discovery of the specific location of the protein in the central nervous system after being injected into crickets (Acheta domestica)”. The title seems to be broad than the data presented in the manuscript. Using biochemical approach’s, the authors propose Sol g 2.1 recombinant protein and its localization at the CNS in fire ants and its role in paralysis. The techniques used in the manuscript are up to date and the experiments are well performed. However, I found certain things which need clarification and attention to strengthen the conclusion.

Title: Please re frame the present title which is confusing and complex.

Abstract: The abstract need to be modified with the major highlights included in the manuscript. In the present form it’s not clear.

Introduction:

Please update the introduction with latest references (last 5-7 years) which are missing. The last paragraph of the introduction needs some modifications which can provide a jest of the manuscript.

Material & Method:

What is the main reason to use the adult male crickets?

How do you setup the concentrations that need to be injected to crickets? How do you standardize it? What are the minimal concentrations were insect gets paralyzed? Do you perform bio assays before setting up the experiments? Please provide some details in the text.

Please provide Procedure of extraction of venom and how do you prepare venom concentrations which is missing in the text?

Insect rearing conditions need to be explained in brief like temperature and humidity etc.

Discussion: This section is poorly written and can be surely improved with highlights to the key findings points and their supporting appropriate references. Please provide some details on the mechanism and metabolic pathways which is missing.

General points:

1.     Please check the typos throughout the manuscript.

2.     The details of the manuscript needs more clarity which is missing in the present form. Please go for English correction with a native speaker of a professional company.

Conclusive remarks: The manuscript contains interesting and significant findings. It still needs some corrections (typos, italics and grammatical errors) which is very important. I still ask the authors to rearrange and extend information on certain parts of the manuscript, especially about Material and method, and discussion with additional references. I do think that the manuscript contains important issues, information, interesting approaches, and techniques, which can lead to a proper understanding the role of Sol g 2.1 in paralysis. So, I consider this manuscript suitable for the publication after the suggested clarifications and inputs in IJMS.

Please go for English correction with a native speaker of a professional company.

Author Response

We attached the response file here.

Reviewer 2 Report

This manuscript describes various investigations concerning to the insect paralysis promoting activity of Sol g 2.1, one of the proteins in the fire ant venom.

Experiments include preparation of the recombinant protein, spectroscopic and computational stereo-structure analyses, paralytic activity assay, immunohistochemistry analyses, and protein-protein interaction modeling.

Based on the results of each of these analyses, Sol g 2.1 is proposes to act as a transporter of the hydrophobic paralytic compounds to the octopamine receptor on the nervus system.

 Although many experiments have been conducted, the relationships among them are clearly described, and the discussion and the final hypothesis of the activity mechanism are easy to understand. If bioactive substances can be transported effectively, it is expected to be a promising pest control strategy.

 It needs further studies for the final conclusion and practical application, there is enough data available to support the hypothesis at this time.

 However, there are a few missing parts, and we believe that adding these parts will make it possible to accept for publishing.

 1.       The design of the recombinant protein and its expression and purification should be included briefly in the result part. This will make the Table 1 easier to understand.

 2.       Negative control is necessary for immunohistochemistry images to see the background, non-selective signals, and noise. Negative control can be a supplement.

 3.       Figure 5B is difficult to see. Same level signals are observed in the marker. It is better to reduce the standard material, and stain longer to clarify the sample signals.

Author Response

We attached the response file here.

Reviewer 3 Report

The insect paralytic peptides (PP) or  growth blocking peptides (GBP) are known but the relationship of the current issue to them is not clear. Association between PP and dopamine has been studied but with tyramine as disscussed here. The authors should clarify these points or otherwise it is confusing. Enlarged  histological pictures with a clear reference with neuroendocrinological landscape is needed. 

Author Response

We attached the response file here.

Round 2

Reviewer 1 Report

Reviewer #:

Now, this paper is exceptionally well-conceived and well-written. The author has skillfully integrated the specifics into the current revised version of the manuscript. Please ensure that the incorporation are revised meticulously. I have identified some typos in the integrated text; kindly rectify them. Lastly, please conduct a thorough review of the references to confirm their adherence to the IJMS format. With these modifications, I endorse the publication of this article in IJMS."

Author Response

We attached the "Response for reviewer 1 (2nd revised)" file here.

Thank you for your kind suggestions.

Reviewer 3 Report

I addressed a confusion in the noenclature and functions here. "Paralytic peptide" has been used for different species and different functions and the authors must clarify in Introduction.

The figures showing the profile of cricket body are grasshopper's. No body take the current figures as cricket lateral aspect. 

Corrections are needed in some places but I had no problem to read the MS.

Author Response

We attached the "Response for reviewer 3 (2nd revised)" file here.
